# ViStruct: Visual Structural Knowledge Extraction via Curriculum Guided Code-Vision Representation

**Yangyi Chen[1], Xingyao Wang[1], Manling Li[2], Derek Hoiem[1], Heng Ji[1]**
[1] University of Illinois Urbana-Champaign [2] Northwestern University
{yangyic3, xingyao6, manling2, dhoiem, hengji}@illinois.edu

## Abstract

State-of-the-art vision-language models (VLMs) still have limited performance in structural knowledge extraction, such as relations between objects. In this work, we present ViStruct, a training framework to learn VLMs for effective visual structural knowledge extraction. Two novel designs are incorporated. First, we propose to leverage the inherent structure of programming language to depict visual structural information. This approach enables explicit and consistent representation of visual structural information of multiple granularities, such as concepts, relations, and events, in a well-organized structured format. Second, we introduce curriculum-based learning for VLMs to progressively comprehend visual structures, from fundamental visual concepts to intricate event structures. Our intuition is that lower-level knowledge may contribute to complex visual structure understanding. Furthermore, we compile and release a collection of datasets tailored for visual structural knowledge extraction. We adopt a weakly-supervised approach to directly generate visual event structures from captions for ViStruct training, capitalizing on abundant image-caption pairs from the web. In experiments, we evaluate ViStruct on visual structure prediction tasks, demonstrating its effectiveness in improving the understanding of visual structures. The code is public at https://github.com/Yangyi-Chen/vi-struct.

## 1 Introduction

Vision-language models (VLMs) exhibit impressive performance across various multimodal downstream tasks (Li et al., 2019a; Tan and Bansal, 2019; Chen et al., 2023b; Wang et al., 2022a), such as visual question answering (Antol et al., 2015; Yuan et al., 2023), image captioning (Karpathy and Fei-Fei, 2015), visual event extraction (Li et al., 2022b; Wang et al., 2022c), chart understanding (Zhou

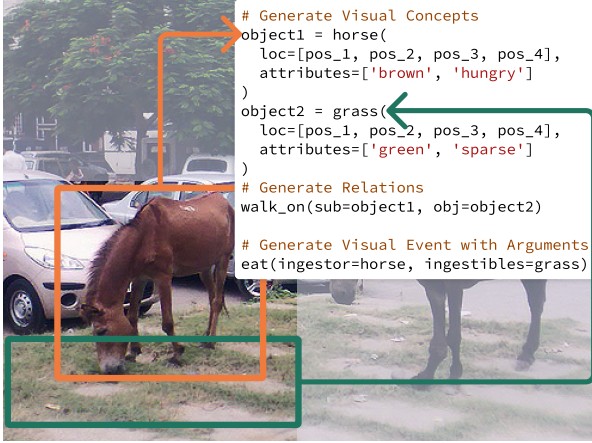

```python
# Generate Visual Concepts
object1 = horse(
    loc=[pos_1, pos_2, pos_3, pos_4],
    attributes=['brown', 'hungry']
)
object2 = grass(
    loc=[pos_1, pos_2, pos_3, pos_4],
    attributes=['green', 'sparse']
)
# Generate Relations
walk_on(sub=object1, obj=object2)

# Generate Visual Event with Arguments
eat(ingestor=horse, ingestibles=grass)
```

Figure 1: Programming language enables a unified representation of visual structural information in code-vision representations, including concepts, attributes, locations, relations, and visual events.

et al., 2023), and multimedia misinformation detection (Fung et al., 2021; Luo et al., 2021). The recipe lies in the pre-training on web-scale data, enabling substantial enhancements in vision-language representation (Radford et al., 2021; Li et al., 2022a).

However, recent evaluations expose the limitations of existing VLMs models in capturing visual structural information (Hendricks and Nematzadeh, 2021; Zhao et al., 2022; Yüksekgönül et al., 2022), rendering them ineffective for visual structural knowledge extraction. For example, Yüksekgönül et al. (2022) argue that VLMs represent images as bag-of-words, disregarding relations among objects. In this work, we present **ViStruct**, a training framework for VLMs to improve their visual structural knowledge extraction ability. There are two key challenges in multi-granularity visual structural knowledge learning: (1) capturing the structures at each granularity, for which we propose code-vision representations, and (2) recognizing the dependencies between different granularities, for which we propose a curriculum pyramid to guide the knowledge learning.

The motivation of employing code-vision rep-

resentations is to mitigate the negative impact of commonly employed techniques such as contrastive learning (Radford et al., 2021) and image-conditioned language modeling (Li et al., 2023), which limit VLMs to focus solely on primary objects, resulting in "bag-of-words" representations. we draw inspiration from prior successful application of programming language in representing linguistic structures (Madaan et al., 2022; Wang et al., 2023b). We propose to extend this "code for structure representation" principle to multimodal settings for explicit representation of visual structures. Initial attempts to employ code representation in visual reasoning (Gupta and Kembhavi, 2023; Surís et al., 2023) focus solely on object information and lack perception abilities, but relying on the off-the-shelf object detectors as tools. As a result, we establish a multi-granularity code representation framework to capture structured visual information, including visual concepts, attributes, relations, and events (see Figure 1).

To further capture the interdependencies among various granularities, we propose the curriculum guided visual structural knowledge learning. Our intuition is that foundational knowledge (such as concepts) may enhance comprehension of more complex visual structures (such as visual events). Thus, unlike previous work that involves pre-training VLMs on a mixture of tasks, we build a visual structural knowledge pyramid (see Figure 2), aligning with the pedagogical strategy of progressively addressing simpler to more complex tasks during training (Soviany et al., 2022; Wang et al., 2022b). However, in contrast to conventional curriculum learning methods that focus on arranging data samples to facilitate training for a specific task (Jiang et al., 2014a; Spitkovsky et al., 2010b), our objective is to enable VLMs to understand visual structures comprehensively at all levels. To accomplish this, we incorporate a replay buffer mechanism that selectively reserves a small amount of data at each level to preserve lower-level knowledge throughout the training process.

Moreover, we curate an extensive collection of datasets for ViStruct, called **ViStruct Suite**. This suite includes multi-level structural knowledge, aligned with the curriculum pyramid. We first collect existing labeled datasets and large-scale unlabeled image-caption pairs. Then, we adopt a weakly-supervised approach to facilitate the creation of visual event structure labels from captions

in a scalable manner. To ensure consistent class label definitions, we automate the alignment of all concepts from various datasets to WordNet (Miller, 1995) and FrameNet (Baker et al., 1998) synsets, with minimal manual assistance.

For downstream task evaluation, we find that ViStruct consistently outperforms various baselines on three downstream structure prediction tasks (visual relation detection, scene graph classification, and situation recognition). In addition, we find that code representation and curriculum learning both provide consistent gains on downstream tasks and demonstrate their effectiveness on zero-shot visual structure prediction.

Parsing visual data into various levels of detail has been a longstanding problem. To the best of our knowledge, we are the first to extract multi-granularity visual structured knowledge via code-vision representation learning, which effectively captures the hierarchy among different levels of visual knowledge, offering a comprehensive and compositional understanding of overall structure.

## 2 Related Work

### 2.1 Vision-Language Models (VLMs) for Visual Knowledge Extraction

Visual knowledge extraction has been long optimized as separate tasks for each granularity of extraction, such as object detection, relation detection, scene graph parsing, situation recognition and event extraction, as further detailed in Appendix B. Recent years witness great success through VLMs to obtain knowledge of visual concepts (Sadhu et al., 2021) and events (Wen et al., 2021; Deshpande et al., 2022; Li et al., 2022c). The primary goal of VLMs is learning representations from multiple modalities to capture visual details and their alignment with text semantics (Gan et al., 2022; Uppal et al., 2022; Wang et al., 2022c), and the corresponding research can be broadly classified into three stages.

The initial stage of VLMs primarily relies on off-the-shelf object detectors to represent visual modality (Li et al., 2019b; Tan and Bansal, 2019; Lu et al., 2019; Li et al., 2020a) and learn the alignment across modalities (Li et al., 2020c, 2021b).

The second phase of VLMs aims to eliminate the use of resource-intensive object detectors and address the issue of unseen objects (Dou et al., 2022; Huang et al., 2020; Kim et al., 2021; Huang et al., 2021; Xu et al., 2021; Yang et al., 2022; Yao et al.,

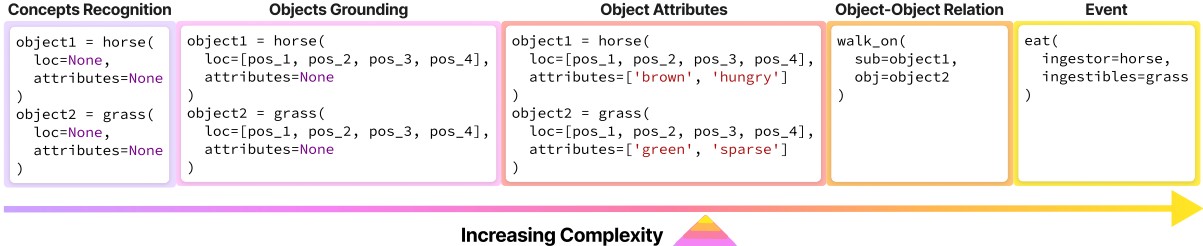

Figure 2: The curriculum pyramid in ViStruct that incorporates five different levels of multimodal knowledge acquisition, progressing from basic to more advanced stages. All levels of visual structures can be uniformly represented using the programming language. The code is generated based on the image in Figure 1.

2022). Large-scale multimodal data is required and emphasized to better align two modalities (Li et al., 2021a; Radford et al., 2021; Jia et al., 2021).

The current stage of VLMs moves towards a more unified pre-training goal to perform various tasks without task-specific adaptations (Wang et al., 2021, 2022a; Li et al., 2023). Some research leverages programming language (Gupta and Kembhavi, 2023; Surís et al., 2023), but lacks perception ability and relies on pre-trained object detectors. Although we witness great success of existing VLMs on benchmark evaluations, in-depth diagnostic investigations have exposed their shortcomings in fundamental visual concept identification (Hendricks and Nematzadeh, 2021; Zhao et al., 2022; Yüksekgönül et al., 2022), raising doubts about whether current VLMs are overemphasizing objects to represent images while ignoring other visual structures, akin to the initial stage VLMs, making them inadequate in structural knowledge extraction. Thus, in this work, we design a training framework to improve VLMs' visual structural knowledge extraction.

## 2.2 Curriculum Learning

Curriculum learning has proven to be a useful technique for improving the learning process by gradually exposing the learner to more challenging examples (Bengio et al., 2009; Karras et al., 2018; Wang et al., 2020; Su et al., 2021; Platanios et al., 2019; Yu et al., 2022). However, it has been studied mainly in a more limited form, ordering and presenting examples to train a single task (Bengio et al., 2009; Spitkovsky et al., 2010a; Kumar et al., 2010; Jiang et al., 2014b; Graves et al., 2017). In this work, we propose the first task-level, multifaceted, multimodal approach to curriculum learning for visual structural knowledge learning.

## 3 ViStruct

We propose ViStruct that learns a vision-language model that takes images as input and generates a set of structural knowledge, such as concepts, relations, and events involved in the images (see Figure 2). We first define a code-vision representation (Sec. 3.1), which serves as the output to decode visual structural knowledge as code blocks. We then propose a curriculum pyramid to guide the training objectives from low-level towards high-level semantics (Sec. 3.2). To train the model, we construct a comprehensive multi-granularity dataset ViStruct Suite (Sec. 3.3), and optimize the model only on information that describe relevant visual strcutural information (Sec. 3.4).

## 3.1 Code-Vision Representation

Visual structural information depicts both visual knowledge elements (such as visual concepts, locations, attributes) and complex structural connections between them (such as relations, and visual events with associated arguments). To encode visual structural knowledge, the first question is to define the appropriate structured representation. Such a representation is required to not only have robust capabilities for illustrating structures and their intricate interconnections, but also be unified and consistent across multiple levels of semantic granularity. It is also expected to be adaptable and extendable to accommodate other future semantic granularities. In order to address these requirements, we leverage the inherent capabilities of programming language (PL). We adopt Python as the PL in this paper due to its straightforwardness and popularity, allowing us to effectively harness its structured representation power and well-resourced data for code-vision pretraining. Similar to PL, we define ViStruct including both class definition and

| Visual Structural Information & Definition | PL class definition |
|---|---|
| Visual structure | Class name |
| Event arguments/subject (object) in relation | Arguments for class instantiation |
| Human-written definition | Docstring |

Table 1: The alignment of visual structures, visual semantic roles, annotated definitions, and programming language (PL) class definitions. The concrete examples are shown in Figure 3.

**Definition of Visual Concept**

```
class horse(Object):
    """
    Definition: solid-hoofed herbivorous quadruped
    domesticated since prehistoric times
    """
    def __init__(loc=None, attributes=None):
        super().__init__(loc=loc, attributes=attributes)
```

**Definition of Visual Event**

```
class building(Activity):
    """
    Definition: This frame describes assembly or
    construction actions, where an Agent joins
    Components together to form a Created_entity.

    agent: The Agent builds a Created_entity.
    created_entity: Created_entity identifies
    the entity that is created in building.
    """
    def __init__(agent=None, created_entity=None):
        self.agent = agent
        self.created_entity = created_entity
```

Figure 3: Examples of the class definitions of basic visual concepts and visual events.

instantiation of visual structural information.

**Class Definition of Visual Structural Information.** To define a general set of multi-granularity visual structures, we combine two external knowledge resources: WordNet (Miller, 1995) for basic visual concepts and relations; and FrameNet (Baker et al., 1998) for visual events. Class definitions in PL provide a unified and structured way to represent the visual structures, semantic roles, and human-written definitions in WordNet and FrameNet. Table 1 details the structure-class mapping we defined. For each visual concept, we define object location `loc` for object grounding, and `attributes` for object properties, as shown in the "horse" example in Figure 3. On top of that, we define visual events, which have visual concepts as participants, characterized by their semantic roles (such as `agent` and `create_entity` in the example visual event "building" in Figure 3). As discussed in Sec. 3.4, these definitions are used to warm up the model through finetuning.

**Instantiation of Visual Structural Information.** As shown in Figure 1, the multi-granularity visual structural information of each image can be conve-

**Algorithm 1** The training algorithm described in Sec. 3.4. We omit the warm-up process for brevity.

1: $P_\theta$ ▷ a pre-trained vision-language model
2: $D_{\text{stage}} = [D_{\text{CR}}, D_{\text{OG}}, D_{\text{OA}}, D_{\text{OR}}, D_{\text{E}}]$ ▷ Stage-specific datasets defined in Sec. 3.2
3: $D_{\text{RB}} = \{\}$ ▷ Replay buffer in Sec. 3.2
4: **for** $D_s$ in $D_{\text{stage}}$ **do**
5:      $D_{\text{train}} = D_s \cup D_{\text{RB}}$
6:      Finetune $M$ using $D_{\text{train}}$
7:      $D_{\text{RB}} = D_{\text{RB}} \cup \text{random\_subset\_of}(D_s)$
8: **end for**

niently represented in a unified and structured manner. In this work, we consider the most salient visual structural information, including concepts, locations, attributes, relations, and visual events with associated arguments. The structure-code mapping is detailed in Table 2. It is noteworthy that our code-vision representation can be effortlessly expanded to encompass supplementary visual details, such as affordance, object states, and similar aspects.

### 3.2 Curriculum-Based Training for Knowledge Extraction with Replay Buffer

Existing vision-language pre-training approaches are mostly based on end-to-end training on a mixture of tasks (Li et al., 2023), which operates differently from the way humans learn. In human learning, for example, the well-validated Montessori method (Lillard et al., 2017) focuses on a curriculum of self-directed learning in stages, where demonstrated mastery of elementary concepts is required before advancing to those more complex. Based on this insight, we propose a curriculum pyramid that directs VLMs towards progressively learning to extract visual structural knowledge from simpler to more complex concepts. In particular, we introduce the replay buffer mechanism to prevent the forgetting of foundational structural knowledge obtained at the lower levels of the curriculum. The training algorithm is described in Alg. 1.

**Curriculum Pyramid.** We present our curriculum pyramid in Figure 2. We categorize the process of learning complete visual structural knowledge into five levels, based on the assumption that knowledge acquisition at lower levels (such as concepts) can facilitate the acquisition of higher-level knowledge (such as visual events). For example, in Figure 1, identifying the "horse" and "grass" concepts contributes to understanding the "eat" event with

| Structural Information | Python Code Representation |
|---|---|
| Concept | `object = <object_synset>(loc=None, attributes=None)` |
| Grounded Object | `object = <object_synset>(loc=[pos_1, pos_2, pos_3, pos_4], attributes=None)` |
| Attribute | `object = <object_synset>(loc=[pos_1, pos_2, pos_3, pos_4], attributes=[<attribute_synset>])` |
| Relation | `<relation_synset>(sub=<defined_object>, obj=<defined_object>)` |
| Event | `<event>(<argument_name>=<object_name>, <argument_name>=<object_name>)` |

Table 2: The correspondence between visual structural information and Python code representations.

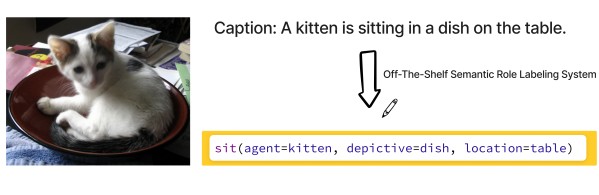

Figure 4: We generate the visual events with argument roles from the captions using the off-the-shelf system.

these two participants.

Conditioned on an image input, we train the model to reconstruct masked visual structures represented using code (e.g., arguments). Specifically, we mask out targeted visual information in the code and treat this masked sequence along with the image as input to the model. For example (Figure 1), in object attributes learning stage, we mask out the target attributes: `object = horse(loc=[pos_1, pos_2, pos_3, pos_4], attributes=[<mask>])`. Then we train the model to generate the code sequence replacing the `<mask>` with "brown" and "hungry". Formally, given an image $i$, the code sequence $s$ and a mask function $\text{mask}(\cdot)$ to mask different information for different stages. We use an auto-regressive loss to train the code-vision representation model on the dataset $D$, that is, we are minimizing the following loss function:

$$\mathcal{L}(\theta) = - \sum_{(i,s) \in D} \log P_\theta(s|i, \text{mask}(s))$$

where $\theta$ refers to the model parameters, $D$ is the dataset we used for training, which composed of a stage-specific dataset $D_{\text{specific}} \in \{D_{\text{CR}}, D_{\text{OG}}, D_{\text{OA}}, D_{\text{OR}}, D_{\text{E}}\}$ for all the five stages in Figure 2 and a replay buffer $D_{\text{RB}}$, that is, $D = D_{\text{specific}} \cup D_{\text{RB}}$ for each stage. We use Figure 1 as a running example to introduce each stage in the pyramid.

The initial stage involves basic **Concept Recognition (CR)**, which applies VLMs to identify the most prominent objects in the images presented (e.g., horse and grass). In a recent study by Kamath et al. (2022); Whitehead et al. (2021), it is demonstrated that decoupling the acquisition of basic concepts from the learning of task-specific skills is effective. We adopt this approach and anticipate that VLMs will gain a range of fundamental visual concepts, including rare object categories, at this level. At the second level, VLMs are required to perform **Object Grounding (OG)**, which involves the identification of objects with specific, concrete locations in the images. The third level places emphasis on **Object Attribute (OA)** detection, with the goal of inferring the attributes associated with particular objects (e.g., the horse is brown and hungry).

The initial three levels facilitate the *precise understanding of objects*. We then comprehend the interaction *between objects* and the *deep semantics* embedded within the images. The fourth level centers on visual **Object Relation (OR)** detection, with the objective of identifying the basic relations between two objects (e.g., predict a *walk-on* relation between horse and grass). In the final level, the focus is on high-level comprehension of the images such as **Event (E)**. Beyond the perceptual tasks, visual event detection requires VLMs to provide a summary of what is happening in the images and the participants involved in the event (e.g., the horse is eating grass).

We highlight the flexibility of our curriculum pyramid that can be easily extended to incorporate new structural information at different levels with the goal of improving overall multimodal knowledge acquisition. This adaptability is facilitated by the flexible code-vision representations employed.

**Replay Buffer.** We aim to empower VLMs with the capacity to proficiently acquire and maintain structural knowledge across various levels, aligning with the underlying motivation of lifelong learning research (Biesialska et al., 2020; Wang et al., 2023a). Although the acquisition of lower-level knowledge remains vital for higher-level knowledge attainment, directly training VLMs on high-level tasks can lead to catastrophic forgetting, especially when dealing with infrequent and rare concepts. To prevent this, we build the replay buffer by random sampling at the level of the curriculum pyramid. During training, VLMs utilize the

data samples specific to their respective level (i.e., samples from previous stages stored in the replay buffer). The amount of data we sampled from each stage can be found in Table 7 in the Appendix.

## 3.3 ViStruct Suite Construction

We assemble a comprehensive dataset collection consisting of both labeled and unlabeled datasets for ViStruct corresponding to each level in the curriculum pyramid. The selected datasets along with their statistics are listed in Table 7 in the Appendix. To curate the object recognition dataset, we exclude rare classes with less than 500 samples and retain a total of 10,450 categories. On top of it, we add object grounding with 80.6M categories, object attributes with 14.6M attribute annotations, and object relations, using the resources in Table 7. For visual event understanding, we propose a weakly-supervised method to generate training data from the large-scale image-caption pairs in Table 7. As shown in Figure 4, for each image, we leverage the off-the-shelf semantic role labeling model[1] to perform semantic parsing on the paired caption. Aligning with ViStruct, we adopt the FrameNet for the event and corresponding argument definitions. More details are included in Appendix E.

We gather the aforementioned datasets and conduct a subsequent post-processing procedure. To establish consistency with ViStruct across various datasets, we align all categories from different datasets to the WordNet and FrameNet synsets through the automatic mapping process implemented in the NLTK toolkit (Bird et al., 2009). In cases where multiple categories in a dataset are assigned to the same synset, we manually adjust the mapping based on our prior knowledge.

## 3.4 Model Training

ViStruct is based on continued pretraining, and can work on any pre-trained VLMs. In this work, we adopt OFA-base (Wang et al., 2022a) as our backbone model since it offers a simple seq-to-seq modeling approach. We train the model by feeding images $i$ and the masked sequence $mask(s)$ as defined in Equation 3.2 as input to the encoder, and train the decoder to produce $s$ conditioned on the encoder outputs. Before the curriculum training, we include a warm-up pretraining stage to facilitate the models' acquisition of fundamental code syntax structures and visual concepts with their definitions (see

---

[1] https://github.com/chanind/frame-semantic-transformer

Appendix C). For the initial three levels in the curriculum, the model is trained for one epoch, while for the last two levels involving deep-semantic understanding, the model undergoes three epochs of training. Specifically, we introduce a focusing optimization trick that only optimize loss for the masked tokens (see Appendix D for more details).

## 4 Experiment

We evaluate ViStruct on three structure prediction tasks, including visual relation detection (Yu et al., 2023), scene graph classification (Zellers et al., 2018; Yao et al., 2021), and situation recognition (Yatskar et al., 2016, 2017). We refer to Appendix A for implementation details.

### 4.1 Visual Relation Detection

**Dataset & Metrics.** We evaluate the visual relation detection task on Visual Genome (Krishna et al., 2017). Following Krishna et al. (2019), we adopt the refined version that only contains 20 well-defined relation categories. Each image is annotated with identified objects and their relations. The task is to *detect the relation* between two given objects (provided along with their bounding boxes).

For evaluation metrics, we measure two types of model performance following previous work (Yao et al., 2021). First, we report the micro-averaged recall in top K relation predictions (R@K). Second, we evaluate the marco-averaged recall of all 20 relation types across the top K predictions (mR@K), intending to measure the performance on long-tail relations. In our study, we select K=50, 100.

**Baselines.** We compare ViStruct against three types of baseline methods. (1) Task-specific models: We consider the Neural motifs (Zellers et al., 2018) (**Motif**) that incorporates task-specific designs (e.g., off-the-shelf object detectors); (2) Extra supervision-enhanced methods: We consider the distant supervision method proposed in Yao et al. (2021), which introduces extra signals for pretraining (**Motif+P**) and denoising (**Motif+D**); (3) Ablations of ViStruct: We aim to measure the separate gains of two designs within ViStruct. Specifically, we quantify the gains of code-vision representations by measuring the performance of the backbone model OFA-base, which is trained on the same datasets but with natural language (i.e., image-conditioned language generation) (**ViStruct-Text**). In assessing curriculum learning ablation,

| Method | R@50 | R@100 | mR@50 | mR@100 | Mean |
|---|---|---|---|---|---|
| Motif* | 67.93 | 70.20 | 52.65 | 55.41 | 61.55 |
| Motif+P* | 73.22 | 75.04 | 60.44 | 63.67 | 68.09 |
| Motif+D* | 76.28 | 77.98 | 60.20 | 63.61 | 69.52 |
| ViStruct-Text | 73.39 | 75.16 | 63.27 | 67.42 | 69.81 |
| ViStruct-Mix | 74.71 | 76.32 | 61.41 | 68.26 | 70.10 |
| ViStruct | **75.71** | **78.53** | **63.71** | **70.71** | **72.17** |

Table 3: The results (%) of the visual relation detection. * denotes results from Yao et al. (2021).

| Method | R@50 | R@100 | mR@50 | mR@100 | Mean |
|---|---|---|---|---|---|
| Motif* | 31.14 | 31.92 | 23.53 | 25.27 | 27.97 |
| Motif+P* | 34.11 | 34.88 | 26.51 | 27.94 | 30.86 |
| Motif+D* | 35.93 | 36.47 | 28.07 | 30.09 | 32.64 |
| ViStruct-Text | 30.55 | 31.04 | 26.26 | 28.73 | 29.15 |
| ViStruct-Mix | 32.34 | 33.65 | 26.22 | 27.87 | 30.02 |
| ViStruct | **34.72** | **35.27** | **30.51** | **32.26** | **33.19** |

Table 4: The results (%) of the scene graph classification. * denotes results from Yao et al. (2021).

we measure the performance of pre-training on the ViStruct Suite mixture (**ViStruct-Mix**).

**Experimental Results.** As shown in Table 3, ViStruct consistently improves the visual relation detection performance regarding both the common and long-tail relation types. The reason lies in the diverse relation types ViStruct encounters during the structural knowledge learning phase, which prevents it from overfitting to some frequent relation types. Additionally, we demonstrate the separate gains of two novel designs in ViStruct.

### 4.2 Scene Graph Classification

**Dataset & Metrics.** We continue to utilize the refined Visual Genome dataset for evaluation purposes. The scene graph classification task involves *predicting object categories and relation types* based on bounding box annotations, thereby assessing the models' concept recognition capabilities besides the ability to detect object interactions. We measure the R@K and mR@K as described in Sec. 4.1, and the correctness of the predicted relation is determined solely by the accuracy of the predicted relation types and object categories.

**Baselines.** We adopt the same baselines Sec. 4.1.

**Experimental Results.** As shown in Table 4, ViStruct consistently outperforms baseline methods. Additionally, the performance gap between ViStruct and baseline methods including task-specific models and two ablations increases compared to the results in Table 3. This can be attributed to (1) the extensive coverage of fundamental visual concepts in ViStruct, aligning with the requirements of scene graph classification for object category identification, and (2) the carefully arranged curriculum that guides the model to progressively comprehend the visual structures, preventing the model from overfitting on the large dataset of visual concepts.

### 4.3 Situation Recognition

**Dataset & Metrics.** We evaluate ViStruct on SWiG dataset (Pratt et al., 2020). Each image in the dataset is annotated with the major activity and corresponding participants (a.k.a., objects). The models must sequentially *predict both the activity and the participants involved*. Following Cho et al. (2022), we measure whether the gold-standard activity is predicted in the top-1 (**Top-1 Predicted Verb**) and top-5 (**Top-5 Predicted Verb**) model predictions. The evaluation of participant prediction entails two aspects: **Value-all** examines whether all predicted participants for an activity are correct, while **Value** evaluates the accuracy of each predicted participant in their assigned role. Participant prediction is considered correct if it matches any of the ground truth annotations. All metrics are calculated on a per-verb-specific basis and subsequently averaged across all verbs.

**Baselines.** We compare ViStruct against three types of baseline methods. (1) Traditional task-specific models: We include GraphNet (Li et al., 2017), CAQ w/ RE-VGG (Cooray et al., 2020), and Kernel GraphNet (Suhail and Sigal, 2019), as studied in Cho et al. (2022); (2) State-of-the-art task-specific models based on Transformers: We consider the CoFormer model that consists of two interactive modules (Cho et al., 2022). (3) Ablations of ViStruct: We consider the same ablations of ViStruct introduced in Sec. 4.1.

**Experimental Results.** As shown in Table 5, ViStruct demonstrates substantial enhancements in both activity and participant detection, which can be attributed to its weakly-supervised training approach utilizing web-scale data. Our approach effectively utilizes the captions to generate visual event structures depicted in the web images. The benefit lies in the increased inclusivity of infrequent events and visual concepts, facilitated by the extensive volume of image-caption pairs. In addi-

| Set | Metric | Top-1 Predicted Verb | | | Top-5 Predicted Verb | | | Ground-Truth Verb | |
|---|---|---|---|---|---|---|---|---|---|
| | Method | verb | value | value-all | verb | value | value-all | value | value-all |
| Dev | GraphNet* | 36.93 | 27.52 | 19.15 | 61.80 | 45.23 | 29.98 | 68.89 | 41.07 |
| | CAQ w/ RE-VGG* | 37.96 | 30.15 | 18.58 | 64.99 | 50.30 | 29.17 | 73.62 | 38.71 |
| | Kernel GraphNet* | 43.21 | 35.18 | 19.46 | 68.55 | 56.32 | 30.56 | 73.14 | 41.68 |
| | CoFormer* | 44.41 | 35.87 | 22.47 | 72.98 | 57.58 | 34.09 | 76.17 | 42.11 |
| | ViStruct-Text | 44.10 | 35.04 | 21.19 | 71.04 | 55.24 | 31.76 | 75.04 | 40.18 |
| | ViStruct-Mix | 44.68 | 35.58 | 21.73 | 71.15 | 55.65 | 32.61 | 75.31 | 40.82 |
| | ViStruct | **46.23** | **38.82** | **24.29** | **74.81** | **60.91** | **37.41** | **79.23** | **46.23** |
| Test | GraphNet* | 36.93 | 27.52 | 19.15 | 61.80 | 45.23 | 29.98 | 68.89 | 41.07 |
| | CAQ w/ RE-VGG* | 37.96 | 30.15 | 18.58 | 64.99 | 50.30 | 29.17 | 73.62 | 38.71 |
| | Kernel GraphNet* | 43.21 | 35.18 | 19.46 | 68.55 | 56.32 | 30.56 | 73.14 | 41.68 |
| | CoFormer* | 44.41 | 35.87 | 22.47 | 72.98 | 57.58 | 34.09 | 76.17 | 42.11 |
| | ViStruct-Text | 44.94 | 35.86 | 21.99 | 71.19 | 55.70 | 32.69 | 75.29 | 40.83 |
| | ViStruct-Mix | 44.72 | 35.60 | 21.77 | 71.03 | 55.51 | 32.62 | 75.23 | 40.81 |
| | ViStruct | **46.89** | **38.58** | **24.91** | **74.57** | **60.40** | **37.53** | **78.99** | **46.62** |

Table 5: The results (%) of the situation recognition task. * denotes results from Cho et al. (2022).

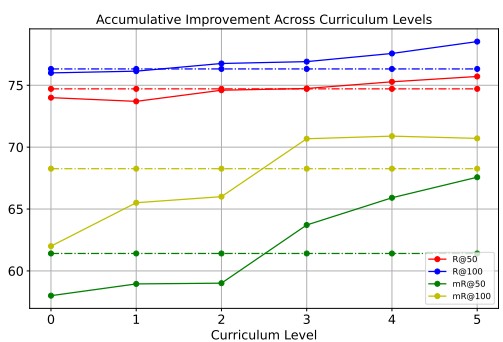

Figure 5: The effectiveness of the curriculum learning framework for visual relation detection. The metrics are introduced in Sec. 4.1. The dotted lines denote the results of ViStruct-Mix.

tion, we observe the concrete gains associated with the utilization of curriculum learning. In alignment with the conclusion in Sec. 4.2, curriculum learning aids the VLMs to focus on high-level visual structure comprehension in the later stages, while simultaneously mitigating the risk of overfitting.

# 5 Further Analysis

## 5.1 Curriculum Learning

We conduct an ablation study to measure the performance of models trained at each stage of the curriculum on the visual relation detection task. We also compare the results of directly pre-training on the mixture of ViStruct Suite (ViStruct-Mix). As shown in Figure 5, as the training progresses, the evaluation metrics regarding two aspects continue to increase, gradually surpassing the performance of ViStruct-Mix, especially for the long-tail rela-

| Method | R@50 | R@100 | mR@50 | mR@100 | Mean |
|---|---|---|---|---|---|
| ViStruct-Text | 7.05 | 10.23 | 3.02 | 4.43 | 6.18 |
| ViStruct | 18.22 | 24.43 | 6.66 | 8.89 | 14.55 |

Table 6: The results (%) of the zero-shot visual relation detection task.

tion types. This demonstrates the benefits of the curriculum learning paradigm for generalization in low-resource settings.

## 5.2 Zero-shot Capability

We evaluate ViStruct on the zero-shot visual relation detection task. The results are presented in Table 6. We compare with the results of training on natural language (ViStruct-Text) introduced in Sec. 4.1, and show that the code-vision representation exhibits significantly better zero-shot performance (i.e., 2.4x better on average). The code-vision representation naturally facilitates the zero-shot visual structure prediction ability, owing to the unified format in both the pre-training and the downstream tasks.

# 6 Conclusion

We present a visual structural knowledge extraction framework ViStruct as a new step towards multi-granularity semantic parsing of visual data. First, we utilize the inherent structured of programming language to represent visual structural knowledge of different levels in a unified way. To capture the interdependencies among various granularities, we then leverage linguistic cues and propose a curriculum learning pyramid

that guides the models to effectively acquire visual structural knowledge of each level. Results show significant and consistent improvements on structure prediction tasks across all levels.

## Limitations and Future Work

This paper utilizes the programming language to represent visual structures, with a focus on knowledge representation. However, due to the constrained capability of the base model, direct code generation is not feasible, resulting in reduced performance in downstream tasks. One potential solution is to integrate more advanced vision-language models, such as BLIP-2 with more capacity. By doing so, the trained models can effectively generate code representations of visual structures for diverse images in downstream tasks.

## Acknowledgement

We thank the anonymous reviewers for their suggestions and comments. This research is based upon work supported by U.S. DARPA ECOLE Program No. HR00112390060 and U.S. DARPA KAIROS Program No. FA8750-19-2-1004. The views and conclusions contained herein are those of the authors and should not be interpreted as necessarily representing the official policies, either expressed or implied, of DARPA, or the U.S. Government. The U.S. Government is authorized to reproduce and distribute reprints for governmental purposes notwithstanding any copyright annotation therein.

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

## A  Adaptation of Downstream Tasks

All evaluations are conducted within the supervised learning paradigm, involving training models using the training dataset, and selecting superior models as dictated by key performance indicators in the validation dataset. The ultimate performance of these models is evaluated on the testing dataset. We enumerate the specifics of adapting to downstream tasks as detailed below.

### A.1  Visual Relation Detection

In this task, the models are provided with ground-truth annotations of the object bounding boxes and their respective categories. The requirement is for these models to effectively predict the visual inter-relations between each pair of objects in the images. In our implementation, each object pair is supplied with their respective annotated bounding boxes, in conjunction with a mask prompt as input parameters for the ViStruct. ViStruct is then tasked with predicting the visual relation between the specified objects. Assuming the pair of objects provided are a horse and grass, the inputs introduced to ViStruct would be as follows:

```
# Generate Concepts
object1 = horse(
        location=[bounding_box],
        attribute=None
        )
object2 = grass(
        location=[bounding_box],
        attribute=None
        )
# Generate Relations
<mask>(sub=object1, obj=object2)
```

Correspondingly, ViStruct is trained to predict the ground truth relation: "walk_on" as output. We meticulously select and refine appropriate verbalizers for every unique relation to ensure that each relation within the label space can be represented by a single, meaningful token in the OFA model.

### A.2  Scene Graph Classification

In this task, the models are provided with the bounding boxes and are required to forecast both the classifications of the objects and their corresponding visual relations. We treat it as a multi-task learning problem, training ViStruct to perform both the object recognition task and the visual relation detection task. For the object recognition task,

we input the mask prompt with annotated location to ViStruct:

```
# Generate Concepts
object1 = <mask>(
        location=
        [bounding_box],
        attribute=None
        )
```

ViStruct is then trained to predict the ground truth object categories, for example, the "horse". The visual relation detection task is trained in the same vain as in Sec. A.1.

### A.3  Situation Recognition

In this task, the models are required to predict the visual event (a.k.a, the activity) in the image with corresponding participants. We treat it as two different tasks, namely the visual event prediction and the event role prediction. For the visual event prediction, we input the prompt as:

```
# Generate Visual Event with
Arguments
```

ViStruct is trained to predict the visual event, for example, "eat". For the event role prediction task, the models are provided with the ground-truth or predictive visual event with a mask prompt as input:

```
# Generate Visual Event with
Arguments eat(ingestor=<mask>)
```

ViStruct is then trained to predict the object for the argument "ingestor". Specifically, in this task, we perform an automatic mapping to convert each potential visual event and object to a single token in the OFA model for training and evaluation. The automatic mapping is based on the word embedding of the OFA model. We select the most similar token, measured by the cosine similarity, with the target verbalizers of visual events and objects.

## B  Visual Structure Learning

Several studies have been conducted consecutively on learning multimodal structural knowledge, commencing with visual concept recognition as a basic task (Deng et al., 2009; Davoodi et al., 2023; Kamath et al., 2022), progressing to object grounding (Yu and Ballard, 2004; Shao et al., 2019; Krasin et al., 2017), object attribute detection (Chen et al., 2023a; Patil and Abhyankar, 2023; Bravo et al., 2022), object-object relation detection (Krishna et al., 2016), and finally visual event understanding (Yatskar et al., 2016, 2017; Cho et al., 2022;

| Type | Pretraining Task | Source | #Sample | #B_Sample |
|------|-----------------|--------|---------|-----------|
| Supervised | Object Recognition | ImageNet21K | 1M | 104K |
| | Object Grounding | Object365, OpenImages, COCO | 2.5M | 1M |
| | Attribute Detection | LSA | 0.5M | 0.1M |
| | Relation Detection | OpenImages, Visual Genome | 0.5M | 0.1M |
| Unsupervised | Event Detection | COCO, SBU, CC3M | 4M | - |

Table 7: The datasets included in ViStruct Suite. #B_Sample denotes the number of samples in the replay buffer.

```
# Generate Visual Concepts
object1 = horse(
    loc=[pos_1, pos_2, pos_3, pos_4],
    attributes=['brown', 'hungry']
)
...
# Generate Relations
walk_on(sub=object1, obj=object2)
# Generate Visual Events
eat(ingestor=horse, ingestibles=grass)
```

Figure 6: The focusing optimization trick prioritizes the semantic content of generated code (highlighted portion) while disregarding the code's syntactic structure.

Li et al., 2020b, 2022b). In this work, we design a curriculum learning pyramid that progressively teaches VLMs the aforementioned structural knowledge and assemble a dataset collection that encompasses multi-level structural knowledge.

## C  Warm-up Pretraining

Prior to engaging in multimodal structural knowledge learning, a warm-up stage is implemented to facilitate the models' acquisition of fundamental code syntax structures and visual concepts with their definitions. To accomplish this, we integrate code training data (Kocetkov et al., 2022), code-formulated class definitions of WordNet and FrameNet synsets, and all datasets within the ViStruct Suite, then subject the models to a single training epoch.

## D  Focusing Optimization

Through our analysis, we observe that the code output contains redundant information regarding the syntax structure, which can be quickly learned during the warm-up stage. Consequently, we propose to focus solely on the semantic content of the generated code, meaning that the loss computation will be limited to the semantic content exclusively. As depicted in Figure 6, this approach can direct models to prioritize structural knowledge and accelerate the training process.

## E  Details of ViStruct Suite

The selected datasets along with their statistics are listed in Table 7. For object recognition, we utilize the pre-processed ImageNet-21K (Ridnik et al., 2021). To curate the dataset, we exclude rare classes that consist of less than 500 samples and retain a total of 10,450 categories, each with a sample size of 100. For object grounding, we leverage object detection datasets, including COCO (Lin et al., 2014), OpenImages (Krasin et al., 2017), and Object365 (Shao et al., 2019), which contain 80, 600, 365 categories respectively. For object attribute detection, we adopt the open-vocabulary LSA (Pham et al., 2022) that contains 14.6M attribute annotations in total. For visual object relations detection, we adopt the OpenImages (Krasin et al., 2017) and Visual Genome (Krishna et al., 2016). We adopt the training split of these labeled datasets to avoid data contamination.

For visual event understanding, we propose a weakly-supervised method to generate training data from the large-scale image-caption pairs in COCO (Lin et al., 2014), SBU Captions (Ordonez et al., 2011), and CC3M (Sharma et al., 2018). As shown in Figure 4, for each image, we leverage the off-the-shelf semantic role labeling model[2] to perform semantic parsing on the paired caption. Aligning with ViStruct, we adopt the FrameNet for the event and corresponding argument definitions.

We gather the aforementioned datasets and conduct a subsequent post-processing procedure. To establish consistency with ViStruct across various datasets, we align all categories from different datasets to the WordNet and FrameNet synsets through the automatic mapping process implemented in the NLTK toolkit (Bird et al., 2009). In cases where multiple categories in a dataset are assigned to the same synset, we manually adjust the mapping based on our prior knowledge.

---

[2]https://github.com/chanind/frame-semantic-transformer