# OpenReview forum: "ViStruct: Visual Structural Knowledge Extraction via Curriculum Guided Code-Vision Representation"
_EMNLP/2023/Conference — EMNLP 2023 Main_

### Official Review · Reviewer_YaB4 · 2023-08-06

**Soundness:** 3

**Excitement:**

4: Strong: This paper deepens the understanding of some phenomenon or lowers the barriers to an existing research direction.

**Paper Topic And Main Contributions:**

This paper presents ViStruct, a training framework for Vision-Language Models (VLMs) to learn visual structural knowledge. The proposed framework adopts curriculum learning on code-vision representations for effective multi-granularity visual structural knowledge extraction.

**Questions For The Authors:**

- Question A: To validate the effectiveness of the code-vision representation, the experimental setting of ViStruct-Text would be clear. Does the warm-up pretraining (depicted in Appendix C) is also utilized for ViStruct-Text?
- Question B: For in-depth analysis of curriculum learning, additional experiments would be needed. Have you conducted experiments under the mixed order of curriculum? About the replay buffer, please report the experimental results without the replay buffer. How did you set the size of the replay buffer of each curriculum? Is the random sampling strategy enough to deal with the infrequent and rare concept (L373)?
- Question C: The authors claim the adopted curriculum prevents the model from overfitting on the dataset of visual concepts (L506-510). Could you explain the connection between the experimental results and the overfitting in detail?

**Reasons To Accept:**

- This paper defines a visual structural knowledge pyramid and collects ViStruct Suite including multi-level structural knowledge aligned with the pyramid. This knowledge pyramid and the curated dataset would help to promote future studies.
- This paper devises an elaborated training framework for VLMs to improve their visual structural knowledge extraction ability.

**Reasons To Reject:**

- Even though the curriculum-guided code-vision representation is an interesting idea, this paper lacks an in-depth ablation study on the proposed method. For example, the additional experiments on the diverse order of curriculum and the design of a replay buffer help to convince the effectiveness of the proposed method.
- In ViStruct Suite, annotations in the visual event detection stage might contain noises since they are obtained by the off-the-shelf semantic role labeling system. A detailed analysis of the curated dataset would be needed.

**Reproducibility:**

4: Could mostly reproduce the results, but there may be some variation because of sample variance or minor variations in their interpretation of the protocol or method.

**Reviewer Confidence:**

4: Quite sure. I tried to check the important points carefully. It's unlikely, though conceivable, that I missed something that should affect my ratings.

**Typos Grammar Style And Presentation Improvements:**

- L413: continued pertraining
- L463: for pertaining

---

> ### Author Rebuttal · Authors · 2023-08-28
>
> Thank you for taking the time to review our paper and for providing invaluable feedback. We sincerely appreciate the insights and recommendations you've provided.
>
> # Reason to Reject
> **R1: Lacks an in-depth ablation study on the proposed method. For example, the additional experiments on the diverse order of curriculum and the design of a replay buffer help to convince the effectiveness of the proposed method.**
>
> A: Thanks for pointing it out! For the design of the curriculum, we do not try enumerating all diverse orders to finalize the curriculum in our work. The design is based on two factors: (1) Empirical evidence drawn from human learning; (2) A certain amount of trail and error in the preliminary experiments. We do not present the experimental results of (2) in our paper because we can not draw reliable conclusions from the obtained results due to the variance. For example, we have tried: (a) first performing the object relation detection task before the concept recognition task; (b) reversing the curriculum order due to the findings that reverse curriculum (from hard to easy) may contribute to the improvement. (Considering the best performance we observed in these two designs) Both of them achieve worse performance compared to directly training on the mixture of datasets (ViStruct-Mix) without curriculum guidance. We will discuss these in the next revision.
>
> In reference to the replay buffer's design, our initial approach did not include this strategy. Consequently, for the average performance, we observed results of 69.42% for visual relation detection and 29.87% for scene graph classification tasks. Notably, these outcomes were even less favorable than those from our other two ablations. It was these findings that motivated us to incorporate the replay buffer into our methodology. We will discuss these in the next revision.
>
>
>
>
>
> **R2: A detailed analysis of the curated dataset would be needed.**
>
> A: Thanks for pointing it out! Yes, we acknowledge that the curated visual event detection dataset does contain noise because the off-the-shelf semantic role labeling system is not perfect. Our strategy is increasing the dataset size of visual event detection to mitigate the noise, which aligns with the common strategy adopted in pretraining on web-scale datasets [1,2]. Specifically, we utilize a mix of COCO, SBU, and CC3M that contains around 3M examples to create our visual event detection dataset. In the author response period, we conduct a human evaluation of our automatically generated visual event detection dataset. We sample 100 samples from the dataset, evaluating the accuracy of (1) visual event, and (2) corresponding argument roles. For the visual event, we find that 93% of samples are correct, 4% of samples are ambiguous (may contain other correct labels), and only 3% are incorrect. For the argument roles, we find that 87% of samples are correct, 8% of samples overlook some entities mentioned in the captions, and 5% of samples contain wrong extracted arguments. We will include these results in the revision.
>
>
> [1] ALIGN: Scaling Up Visual and Vision-Language Representation Learning With Noisy Text Supervision. Chao Jia and Yinfei Yang.
>
> [2] Learning Transferable Visual Models From Natural Language Supervision. Alec Radford, Jong Wook Kim, Chris Hallacy, et al
>
>
>
>
>
>
>
> # Questions
> **QA: Does the warm-up pretraining (depicted in Appendix C) is also utilized for ViStruct-Text?**
>
> A: In our experiments, we have tried this strategy but observed worse results (-0.43 on average) on the evaluated visual relation detection task. So the results of ViStruct-Text we report in our paper are without the warm-up pretraining.
>
>
> **QB: Have you conducted experiments under the mixed order of curriculum? About the replay buffer, please report the experimental results without the replay buffer. How did you set the size of the replay buffer of each curriculum? Is the random sampling strategy enough to deal with the infrequent and rare concept (L373)?**
>
> A: For the mixed order of curriculum and replay buffer, please refer to our response to the first reject reason. We report the size of the replay buffer of each curriculum in Table 7 of our paper. The size is determined to ensure that 100 samples of each concept are maintained in the replay buffer for each dataset (e.g., ImageNet21K, Object365).
>
> Sorry for the confusion about the random sampling strategy. We do not employ a purely random sampling approach across the entire dataset. Instead, our sampling strategy is category-specific. That is, for every category, we randomly select a predetermined number of examples to guarantee an equal representation of both rare and common concepts. We will make it clear in our next version.
>
>
>
>
>
>
> **QC: Explain the connection between the experimental results and the overfitting in detail.**
>
> A: In Table 4, we present a comparison between the performance of our proposed method, ViStruct, and its ablation, ViStruct-Mix, which is directly pretrained on the ViStruct Suite mixture. The results clearly indicate that ViStruct surpasses ViStruct-Mix with an average relative improvement of 10.55%. This significant difference underscores the advantage of curriculum training, suggesting its ability to progressively guide the model in learning complex visual structures in the later stages of training. Directly training on the ViStruct Suite mixture poses challenges, especially given that the concept recognition dataset incorporates a multitude of rare and long-tail visual concepts. Such complexities necessitate significant modeling capacity. Without the guided progression of our curriculum, models risk being overwhelmed by these intricacies, which can impede their ability to learn and represent more complex visual structures like visual events. This, we believe, would increase susceptibility to overfitting, especially on rare concepts.
>
> # Typos
> Thanks for pointing them out! We will correct them in the next version.

---

### Official Review · Reviewer_4qDb · 2023-08-06

**Typos Grammar Style And Presentation Improvements:** None
**Soundness:** 4

**Excitement:**

4: Strong: This paper deepens the understanding of some phenomenon or lowers the barriers to an existing research direction.

**Missing References:**

None

**Paper Topic And Main Contributions:**

This paper studies visual structural knowledge extraction. Concretely, given an image with questions about the visual relations between two objects (or multiple objects), the model expected to provide the correct answer. To study this problem, the paper proposed a very interesting approach that encodes the image into a programming language and leveraged this language for visual relation understanding. The approach could also leverage the pre-trained code based LLM to further improve the performance.

**Questions For The Authors:**

I think this paper is very interesting and exciting. I have no more question besides the one mentioned in the reasons to reject.

**Reasons To Accept:**

The idea of encoding an image using programming language is quite interesting.

This programming language based approach could potentially further achieve improved performance using code based LLM.

The paper converts the visual relation prediction task into a masked language prediction problem. This is very interesting.

**Reasons To Reject:**

I would recommend either add a section describing the inference stage or move the section A from appendix to main text. For several rounds of reading the main text, I couldn't figure out how to do the inference. After reading the appendix, I realized it is doing mask prediction.

I hope the author could provide more implementation details for how to reproducing the results in either main text or appendix. I found it might be a little bit hard to find those information.

I think the author might also want to mention that the program language (generated) is not intended for directly running from some python environment. In the first round of reading, I misunderstood that the generated code is intended for directly running.

**Reproducibility:**

5: Could easily reproduce the results.

**Reviewer Confidence:**

4: Quite sure. I tried to check the important points carefully. It's unlikely, though conceivable, that I missed something that should affect my ratings.

---

> ### Author Rebuttal · Authors · 2023-08-28
>
> We are excited that you like our paper! We thank you for your valuable feedback and suggestions.
>
> 1. **Inference Stage and Section A from Appendix**:
> We understand the importance of clarity in explaining the inference stage, especially given its critical role in the methodology. In response to your suggestion, we will move Section A from the appendix to the main text. This will allow readers to have a seamless understanding of the mask prediction process without the need to refer to the appendix.
>
> 2. **Implementation Details**:
> We apologize for any oversight regarding the provision of implementation details. To ensure reproducibility, we will enhance our paper by adding a more comprehensive section.
>
> 3. **Clarification on Generated Program Language**:
> Thank you for pointing out this potential point of confusion. We will make it explicit in the main text that the generated program language is not intended for direct execution in typical python environments.

---

### Official Review · Reviewer_CE6x · 2023-08-10

**Soundness:** 2

**Excitement:**

4: Strong: This paper deepens the understanding of some phenomenon or lowers the barriers to an existing research direction.

**Paper Topic And Main Contributions:**

This paper proposed ViStruct, a novel curriculum framework for training vision-language models (VLMs) with code representation of visual structural knowledge. It leverages the inherent structure of programming languages and converts visual information to Python-style code which ranges from basic concepts to complex events. Additionally, they employ curriculum-based learning, progressively teaching VLMs to understand varying visual structures, they also collect a new dataset for pre-training. The experimental result shows that the proposed method can output baselines and many other previous works.

**Questions For The Authors:**

See weaknesses.

**Reasons To Accept:**

1. I like the idea of structuralizing the vision information (concept, attribute, location, relation, etc) into codes. Compared with text, codes are more organized and more efficient to express high-level logic.
2. The curriculum learning framework propagates from the basic concept level up to the event level according to the complexity, which is intuitively reasonable. Although almost no new methods are proposed, combining previous methods to adapt to the new problem is still meaningful.

**Reasons To Reject:**

1. Main concern: Only one backbone (OFA-base) is evaluated and the model size is too small. At least two or three different pre-trained VLM backbones are required to demonstrate the generalizability of the proposed method as it's a plug-in method on top of pre-trained VLMs. Also, the model scale is too small (just base). I wonder if the conclusion that curriculum learning helps can still hold with larger pre-trained VLM, i.e., more pre-trained natural text knowledge injected.
2. Too few previous works are compared in Tab1 and Tab2 and they are too outdated.
3. I didn't see any evaluation on grounding and attribute prediction. As they are also one of the curriculum levels, it's necessary to evaluate them.

**Reproducibility:**

4: Could mostly reproduce the results, but there may be some variation because of sample variance or minor variations in their interpretation of the protocol or method.

**Reviewer Confidence:**

4: Quite sure. I tried to check the important points carefully. It's unlikely, though conceivable, that I missed something that should affect my ratings.

---

> ### Author Rebuttal · Authors · 2023-08-28
>
> Thanks for your valuable feedback and comments. Our responses are as follows.
>
> # Reason to Reject
> **R1: Only one backbone (OFA-base) is evaluated and the model size is too small**
>
> A: We agree that evaluating our method based on various VLM backbones can make the work more solid. But we only have limited computing resources in the academic lab. In our work, the pretraining takes around 1 month, including some trials and errors. So it would be very difficult for us to conduct experiments using 2 or 3 VLM backbones.
> In addition, we discuss the model scale in the limitation section of our paper and will do try implementing our method on a more powerful model, like BLIP-2, in future work.
>
> **R2: Too few previous works are compared in Tab1 and Tab2**
>
> A: Thank you for your feedback. We have diligently reviewed relevant and latest papers related to these tasks [1,2,3,4,5].
>
> The primary distinction between the methodologies described in the references and the ones in our experiments pertains to the stages of improvement they target. In our work, we primarily focus on enhancing the vision-language representations in the backbone models. In contrast, the predominant thrust of the studies [1,2,3,4,5] is geared towards refining the training phase. This includes augmentations in the training data [4], refining training objectives [1,2], and adopting novel training paradigms [3,5].
>
> It is important to note that the advancements made in the training stages by the recent work [1,2,3,4,5] can be seamlessly integrated with our methodology. This is based on the premise that our approach augments the visual structural knowledge extraction capability of the backbone model. We will ensure clarity on this matter and provide a more comprehensive discussion regarding related research in our subsequent revision.
>
> [1] Graphical Contrastive Losses for Scene Graph Parsing; Ji Zhang, Kevin J. Shih, Ahmed Elgammal, Andrew Tao, Bryan Catanzaro
>
> [2] The Devil is in the Labels: Noisy Label Correction for Robust Scene Graph Generation; Lin Li, Long Chen , Yifeng Huang, Zhimeng Zhang, Songyang Zhang, Jun Xiao
>
> [3] RelTR: Relation Transformer for Scene Graph Generation; Yuren Cong, Michael Ying Yang, Bodo Rosenhahn
>
> [4] Fine-Grained Scene Graph Generation with Data Transfer; Ao Zhang, Yuan Yao, Qianyu Chen, Wei Ji, Zhiyuan Liu, Maosong Sun, Tat-Seng Chua
>
> [5] Panoptic Scene Graph Generation Jingkang Yang, Yi Zhe Ang, Zujin Guo, Kaiyang Zhou, Wayne Zhang, Ziwei Liu
>
>
>
>
> **R3: Evaluation of grounding and attribute prediction**
>
> A: In the author response period, we conduct experiments on grounding due to the time limits. We compare our approach with task-specific models (MDETR [1]) and the ablation that directly using text representation. We evaluate the methods on RefCOCO and RefCOCO+, and report the Acc@0.5 results on the validation split. The results are shown below. We observe that ViStruct can still outperform baselines in the grounding task.
>
> | Method/Dataset | RefCOCO | RefCOCO+ |
> | :-----| :----: | :----: |
> | MDETR | 86.75 | 79.52 |
> | ViStruct-Text | 87.89 | 80.94 |
> | ViStruct | 88.57 | 82.07 |
>
> [1] Mdetr - modulated detection for end-to-end multi-modal understanding; Aishwarya Kamath, Mannat Singh, Yann LeCun, Ishan Misra, Gabriel Synnaeve, Nicolas Carion.

---

### Official Review · Reviewer_cMsY · 2023-08-13

**Soundness:** 3

**Excitement:**

4: Strong: This paper deepens the understanding of some phenomenon or lowers the barriers to an existing research direction.

**Missing References:**

Some more baselines can be considered for table 3 and 4. for eg:
1. Graphical Contrastive Losses for Scene Graph Generation https://arxiv.org/abs/1903.02728
1. RelTR: Relation Transformer for Scene Graph Generation: https://arxiv.org/abs/2201.11460v3
1. Noisy label correction for robust scene graph generation https://openaccess.thecvf.com/content/CVPR2022/papers/Li_The_Devil_Is_in_the_Labels_Noisy_Label_Correction_for_CVPR_2022_paper.pdf

More baselines can be found from citations of the above papers.(there should be many in this line of work)

**Paper Topic And Main Contributions:**

This paper primarily targets the visual structure prediction/extraction - like tasks where, given an image, the model should extract concepts, objects, their attributes, relations etc. The paper uses “code” for representing visual structure. The paper can be categorized in the line of work that has shown that using “code” for representing structural knowledge can prove to be very useful for large language models (especially in the few shot prompting setting). Even though prior work that uses code representations for natural language (common sense reasoning, event extraction) is an inspiration for this work, there are significant differences for eg. 1. This work targets the multimodal setting which is significantly more challenging and 2. Unlike previous works that use code representations, this work does not use prompting, since prompting VLMs might not be as good as prompting of LLMs. Training techniques (curriculum learning) are designed in this paper for generating visual structure in the form of code.

**Questions For The Authors:**

See weaknesses above, all my questions/concerns are included there.

**Reasons To Accept:**

1. Novel idea: Representing visual knowledge as code is a new idea as per my knowledge, including masked code prediction, and curriculum learning.
1. Scalable w.r.t new/larger/better models: I believe as VLMs improve, this technique should also improve.
1. The ViStruct Suite might be useful for benchmarking/training of future visual structure extraction models.
1. Evaluation is performed on multiple tasks

**Reasons To Reject:**

1. Relatively low performance increase:  Although Table 3 and Table 4 show best results are achieved by Vistruct, the numerical values do not seem to be consistent with the value which are made bold in all columns (some non bold numbers are the highest it seems?). I would request the authors to correct this. Also see my next point regarding baselines considered.
1. Insufficient comparison with baselines: There are numerous methods for scene graph generation/relation prediction which do not seem to be included in Table3 and Table 4. They are for example [1,2,3] and it is possibly many other scene graph generation methods (also see this somewhat old repository for baselines https://github.com/microsoft/scene_graph_benchmark). In light of these advances in SGG, the utility of the current method is questionable. Comparison with all are not necessary, but some latest/recent baselines should be included.
1. Limited datasets used: This work uses visual genome for benchmarking the VRD and SGC tasks. Prior methods routinely used datasets like OpenImages. Benchmarking on an extra dataset can improve this work.
1. Generalizability to different pre-trained models: It seems necessary to study the effect of changing pre-trained models. Currently, table 3 baselines and OFA based model of this paper is not really comparable I believe. I do not have a direct solution to this, but having more pre-trained backbones can improve this work and give the reader more clarity of the performance of the model w.r.t pretrained backbone.
1. Curriculum learning: more details are required to claim performance improvements. for eg. is the current ordering of curriculum necessary? are same computation resources/epochs/data size used for the ablation which compares curriculum based learning with non curriculum based learning?
1. Applications of visual structure: Although there are many papers that tackle the problem of visual structure/scene graph generation (and probably some are better than this method as per the previous point), the utility of generating a scene graph alone is questionable. Instead, applications of using scene graphs might be more interesting to improve this work, for eg. this work cites Yüksekgönül et al., 2022 which shows that VLMs have a bag of words nature. Can the current method extract visual relation ships which can then be used to make progress on ARO task provided by Yüksekgönül et al. Joint training for image-text matching and code completion as done in this work, is a possible direction. This point is largely a suggestion and not exactly a reason to reject.

References
1. Graphical Contrastive Losses for Scene Graph Generation https://arxiv.org/abs/1903.02728
1. RelTR: Relation Transformer for Scene Graph Generation: https://arxiv.org/abs/2201.11460v3
1. Noisy label correction for robust scene graph generation https://openaccess.thecvf.com/content/CVPR2022/papers/Li_The_Devil_Is_in_the_Labels_Noisy_Label_Correction_for_CVPR_2022_paper.pdf

**Reproducibility:**

4: Could mostly reproduce the results, but there may be some variation because of sample variance or minor variations in their interpretation of the protocol or method.

**Reviewer Confidence:**

4: Quite sure. I tried to check the important points carefully. It's unlikely, though conceivable, that I missed something that should affect my ratings.

---

> ### Author Rebuttal · Authors · 2023-08-28
>
> We are glad that you like our paper! We have carefully considered your comments and would like to address them below.
>
> # Reason to Reject
> **R1: Relatively low performance increase.**
>
> A: We will correct the incorrect bold highlighting of the results. For the relatively low performance increase, the visual relation detection task (Table 3) may be too simple for evaluating VLMs. Thus, the gap between various methods is relatively small. The scene graph classification (Table 3) is a more challenging task given the performance of existing methods. Although the absolute improvement of ViSutrct seems not significant, the average relative improvement compared to the previous model is 7.55%. We conduct the student t-test to reevaluate the results and show that our approach shows significant improvement compared to the baselines with p<0.1. In addition, the best-performing baseline model in Table 4 (Motif+D) uses extra augmented data for training, while our approach only relies on the original training dataset.
>
>
> **R2: Insufficient comparison with baselines for the visual relation detection and scene graph classification tasks.**
>
> A: Thank you for your feedback and for pointing out the relevant references [1,2,3]. We have diligently reviewed these papers, along with some of the more recent publications related to these tasks [4,5].
>
> The primary distinction between the methodologies described in the references and the ones in our experiments pertains to the stages of improvement they target. In our work, we primarily focus on enhancing the vision-language representations in the backbone models using continued pretraining. In contrast, the predominant thrust of the studies [1,2,3,4,5] is geared towards refining the finetuning phase. This includes augmentations in the training data [4], refining training objectives [1,2], and adopting novel training paradigms [3,5].
>
> It is important to note that the advancements made in the training stages by the recent work [1,2,3,4,5] can be seamlessly integrated with our methodology. Namely, our method and the related work are complimentary. This is based on the premise that our approach augments the visual structural knowledge extraction capability of the backbone model. We will ensure clarity on this matter and provide a more comprehensive discussion regarding related research in our subsequent revision.
>
>
> [1] Graphical Contrastive Losses for Scene Graph Parsing; Ji Zhang, Kevin J. Shih, Ahmed Elgammal, Andrew Tao, Bryan Catanzaro
>
> [2] The Devil is in the Labels: Noisy Label Correction for Robust Scene Graph Generation; Lin Li, Long Chen , Yifeng Huang, Zhimeng Zhang, Songyang Zhang, Jun Xiao
>
> [3] RelTR: Relation Transformer for Scene Graph Generation; Yuren Cong, Michael Ying Yang, Bodo Rosenhahn
>
> [4] Fine-Grained Scene Graph Generation with Data Transfer; Ao Zhang, Yuan Yao, Qianyu Chen, Wei Ji, Zhiyuan Liu, Maosong Sun, Tat-Seng Chua
>
> [5] Panoptic Scene Graph Generation Jingkang Yang, Yi Zhe Ang, Zujin Guo, Kaiyang Zhou, Wayne Zhang, Ziwei Liu
>
>
> **R3: Limited datasets used. This work uses visual genome for benchmarking the VRD and SGC tasks. Prior methods routinely used datasets like OpenImages.**
>
> A: Thanks for pointing it out! Actually, the OpenImages dataset is included in ViStruct Suite, which is used for pretraining. So, we exclude this dataset for evaluation to prevent the data leakage. For the visual genome considered in our work, we spent great efforts on excluding examples that appear in the evaluation set for fair comparison.
>
>
> **R4: Generalizability to different pre-trained models**
>
> A: We agree that evaluating our method based on various VLM backbones can make the work more solid. But we only have limited computing resources in the academic lab. In our work, the pretraining takes around 1 month, including some trials and errors. So it would be very difficult for us to conduct experiments using 2 or 3 VLM backbones.
>
> **R5: More details are required to claim performance improvements. for eg. is the current ordering of curriculum necessary? are same computation resources/epochs/data size used for the ablation which compares curriculum based learning with non curriculum based learning?**
>
> A: We will add those details in the next revision. For the design of the curriculum, we do not try enumerating all diverse orders to finalize the curriculum in our work. The design is based on two factors: (1) Empirical evidence drawn from human learning [1,2]; (2) A certain amount of trial and error in the preliminary experiments. We do not present the experimental results of (2) in our paper because we can not draw reliable conclusions from the obtained results due to the variance. For example, we have tried: (a) first performing the object relation detection task before the concept recognition task; (b) reversing the curriculum order due to the findings that reverse curriculum (from hard to easy) may contribute to the improvement. (Considering the best performance we observed in these two designs) Both of them achieve worse performance compared to directly training on the mixture of datasets (ViStruct-Mix) without curriculum guidance. We will discuss these in the next revision. For a fair comparison of our approach and ablations, we utilize the same datasets and same epochs for training.
>
> [1] Curriculum learning; Yoshua Bengio, Jérôme Louradour, Ronan Collobert, Jason Weston
>
> [2] Progressive growing of gans for improved quality, stability, and variation; Tero Karras, Timo Aila, Samuli Laine, and Jaakko Lehtinen.
>
>
>
> **R6: Applications of visual structure**
>
> A: Thanks for your insightful and interesting suggestions! In this work, we show that our approach can improve the ability of VLM to perform visual structural knowledge extraction. In addition, we are currently working on exploiting the concrete applications of ViStruct. As you suggested, we also consider the open-domain visual relationship extraction. Besides, we also consider mixing the visual structural knowledge learning in ViStruct with the typical end-to-end pretraining to further leverage the web-scale data to improve performance.
>
>
> # Missing References:
> Thanks for pointing them out! We will add these citations in the revision.

---

### Meta-Review · Area_Chair_dfV8 · 2023-09-19

**Recommendation:** 5

**Metareview:**

All reviewers are generally appreciative of the novel idea of representing visual knowledge as code and using curriculum-based learning to guide the vision-language models training process in different complexity stages. The ViStruct Suite is seen as potentially useful for training and benchmarking future visual structure extraction models.

However, one key remaining concern is the limited number and outdated nature of the pre-trained models used for evaluation. The reviewers also suggest further improvements, such as better exposition, detailed analyses of ViStruct Suite and the curriculum learning design.

Given its novelty and potential impact on future work, all reviewers are pretty excited about this work.

---

### Decision · Program_Chairs · 2023-10-07

**Decision:**

Accept-Main

**Comment:**

All reviewers are generally appreciative of the novel idea of representing visual knowledge as code and using curriculum-based learning to guide the vision-language models training process in different complexity stages. The ViStruct Suite is seen as potentially useful for training and benchmarking future visual structure extraction models.

However, one key remaining concern is the limited number and outdated nature of the pre-trained models used for evaluation. The reviewers also suggest further improvements, such as better exposition, detailed analyses of ViStruct Suite and the curriculum learning design.

Given its novelty and potential impact on future work, all reviewers are pretty excited about this work.